# Differences between Scaly-sided Merganser (*Mergus squamatus*) and Common Merganser (*M. merganser*) feather microstructure

**Donghong Li, Shiyu Zhang, Ran Tian, Yongbin Zhao**⬤*, **Guodong Yi**

College of life sciences, Jilin Normal University, Siping, Jilin, China

* syding@jlnu.edu.cn

## Abstract

The microstructural characteristics of feathers are useful for species identification. In this study, scanning electron microscopy was employed to examine the microstructures of contour feathers, rectrices, and down feathers from both the Scaly-sided Merganser (*Mergus squamatus*) and Common Merganser (*Mergus merganser*). The primary objective was to assess inter-species differences and evaluate the potential of these microstructural characteristics as reliable indicators for distinguishing species. Several microstructural characteristics of feathers exhibited significant variations between the two species. In rectrices, significant variations were observed in the prong length, base length, hooklet number, and prong number of distal barbules. Similarly, down feathers exhibited marked differences in the node number, distance between nodes, internode width, and barbule length of downy barbules. Stepwise discriminant analysis, combined with the leave-one-out cross-validation test, further validated the discriminatory power of all microstructural characteristics. For contour feathers, incorporating the base length into the model achieved a 56.9% correct classification rate. In rectrices, the hooklet number and prong length emerged as key discriminators, with a correct classification rate of 91.3%. Most notably, the barbule length, node number, and distance between nodes of down feathers demonstrated exceptional discriminative capabilities, attaining a perfect 100% correct classification rate. Consequently, the barbule length, node number, and distance between nodes of down feathers, may serve as potentially useful morphological markers for differentiating the Scaly-sided Merganser from the Common Merganser.

## Introduction

The microstructure of bird feathers exhibits significant variation among species whereas maintaining strong stability and interspecific specificity [1]. Chandler first proposed feathers as a core and distinctive external feature of birds, suggesting that their microscopic structure, especially that of barbules, is potentially taxonomically significant

**Data availability statement:** All relevant data are within the manuscript and its Supporting Information files.

**Funding:** This study is funded by the Jilin Provincial Department of Science and Technology (http://kjt.jl.gov.cn), under the program titled "Research on Reproductive Strategies for the Scaly-sided Merganser (Mergus squamatus) in the Changbai Mountain Range" (grant number: 20220101331JC). We declare that no authors received a salary from the funder, and the funder had no role in study design, data collection and analysis, decision to publish, or preparation of the manuscript.

**Competing interests:** The authors have declared that no competing interests exist.

and diagnostically indicative of bird species [2]. A notable example of the practical application of these characters is Roxie Laybourne's work, where she identified the European Starling (*Sturnus vulgaris*) as the cause of an aviation accident by analyzing feathers found in the wreckage [3]. To date, many research findings have demonstrated the significant value of feather microstructural features in the identification and classification of bird species, providing crucial evidence for avian taxonomy research [4–8].

Scaly-sided Merganser (*Mergus squamatus*) has been listed as endangered by the International Union for Conservation of Nature (IUCN) since 2002 [9]. Consequently, there is substantial interest in various aspects of Scaly-sided Merganser, including population size [10–12], habitat selection [13,14], genetic diversity [15], migration connections [16], and breeding ecology [17]. In addition to field observations, molecular biology and isotope tracing are often used to answer questions related to these topics. However, due to the rarity and high vigilance of Scaly-sided Mergansers, direct capture in order to take blood or tissue samples is becoming increasingly challenging. Thus, naturally shed feathers may serve as effective research material. It is noteworthy that Common Merganser (*Mergus merganser*), classified as a species of least concern according to the IUCN Red List, is distinguished as the largest and most widely distributed species within the genus *Mergus* in China [18]. Both species belong to the order Anseriformes, family Anatidae, and genus *Mergus*, which share similar ecological behaviors and exhibit some degree of habitat overlap [19]. This presents a challenge for ecologists, as feather samples, whether molted, found, or from an undetermined locality, may originate from both species. Nonetheless, when conducting molecular biology studies on feathers or performing isotope analysis, it is imperative to meticulously differentiate the feathers of Scaly-sided Merganser from those of other species. The uncontrollable timing of feather collection often leads to significant DNA degradation after sample acquisition. Additionally, naturally shed feathers inherently contain a limited amount of DNA, posing considerable challenges for molecular biology analysis. Therefore, morphological differentiation of feather samples is an incredibly useful tool before sequencing DNA. In the case of both Scaly-sided Merganser and Common Merganser, the macroscopic morphology of their feathers is remarkably similar, making distinguishing between the two species challenging through simple observation alone. Hence, a detailed investigation of the microstructure is necessary for precise identification.

This study aimed to investigate the microstructural characteristics of the contour feathers, rectrices, and down feathers in both the Scaly-sided Merganser and the Common Merganser. A quantitative analysis of feather barbules was performed to identify inter-species differences and to assess whether these microstructural features can serve as reliable markers for species identification.

## Materials and methods

### Sample collection

Feathers of Scaly-sided Merganser were collected from Longhu Mountain, Jiangxi Province, China. A group of more than 20 Scaly-sided Mergansers was observed through binoculars during the field survey. To minimize disturbance, feathers were

randomly collected only after they had left the site. All sampling was conducted in publicly accessible areas outside the core protected zones of the species' wintering range. In these regions, non-invasive scientific activities, including the collection of naturally shed biological materials, are explicitly permitted under existing wildlife conservation laws without the need for special permits.

Additionally, feathers of Common Merganser were obtained from the following sources: (i) an adult female Common Merganser specimen from the collection of Jilin Normal University in Siping, Jilin Province, China; (ii) an adult male Common Merganser carcass provided by a volunteer from Shenyang, Liaoning Province, China; and (iii) multiple feathers of the Common Merganser provided by a volunteer from Fuzhou Senyi Ecological Environment Engineering Co., Ltd. in Fuzhou, Fujian Province, China. This professionally registered entity specializes in wildlife specimen restoration and conservation. Its team includes certified taxidermists, senior ecological engineers, and professionals trained in avian anatomy, all ensuring adherence to scientific collection standards. Consequently, these feathers were collected from corresponding anatomical positions on preserved specimens. However, due to the batch collection method and subsequent mixing during packaging and transportation, the exact number of source individuals could not be determined.

## Sample preparation

A total of 120 feather samples were included in this study, representing a minimum of 5 Scaly-sided Mergansers and 3 Common Mergansers. For each species, 20 feathers of each type, namely flank contour feathers, rectrices, and down feathers, were analyzed. The specific sampling strategies were as follows: For Scaly-sided Merganser, 20 feathers of different types were randomly selected from a mixed sample pool using random sampling. The samples of Common Merganser consisted of two parts: Firstly, 18 feathers (3 feathers of each type per individual, totaling 3 types) were randomly collected from 2 individuals; secondly, 42 feathers (14 feathers of each type, totaling 3 types) were randomly selected from the samples donated by the volunteer. All feather samples were first cleaned with tap water, then washed with a 75% ethanol solution, and allowed to air dry [20]. After drying, two barbs from the intermediate plumulaceous region on each of the left and right vanes chosen per feather. Subsequently, these barbs were affixed to conductive adhesive tape.

## Micrometric measurement and counting

Scanning electron microscopy (G6pure model, Phenom, Netherlands) was used to measure and count the microstructural features of each barb at magnifications ranging from 320× to 2600×. The parameters measured and counted included prong length, base length, inter-barbule distance, number of hooklets, and number of prongs on the distal barbule. Additionally, the barbule length, internode width, distance between nodes, and the number of nodes on the downy barbule were also measured and counted. Each parameter was measured or counted once for each barb, resulting in a total of 80 measurements per parameter for each species. Detailed values can be found in the S1 File, and Table 1 provides the exact measurement details.

Table 1. Definitions of scanning electron microscope measurements of microstructural characteristics.

| Characters | Measurements |
|---|---|
| Prong length | Distance from the attachment point on the node and prong to the tip of the prong. |
| Base length | Distance from the attachment point on the barb to the twisted point in the barbule. |
| Inter-barbule distance | Measured at midsection of barb, distance from base of one barbule to the next distal barbule. |
| Distance between nodes | Distance from the second node to the first node. |
| Internode width | Pennulum width between node. |
| Barbule length | Distance from the attachment point on the barb to the tip of the barbule. |

## Statistical analyses

Mean±standard deviation (SD) of each variable was calculated for each species. Given the small sample sizes, the Mann-Whitney U test was adopted to evaluate differences in the microstructure of barbules between the Scaly-sided Merganser and Common Merganser [4]. For each feather type, stepwise discriminant analysis was implemented to identify variables with the strongest predictive power to distinguish between the two species. All statistical analyses were performed using SPSS version 25.0 software (IBM, USA). Statistical significance was defined as P<0.05.

## Results

### Contour feathers

The distal barbules of contour feathers in both the Scaly-sided Merganser and the Common Merganser were composed a base and a pennulum with both hooklets and prongs (Fig 1). Statistical analysis showed that there were no significant differences between the two species in terms of prong length (P=0.087), base length (P=0.064), inter-barbule distance (P=0.869), hooklet number (P=0.312), and prong number (P=0.134) (Table 2).

To delve deeper into the discriminatory potential of these microstructural features, a stepwise discriminant analysis was carried out on each variable. As shown in Table 3, the base length (P=0.02) was the only variable retained in the discriminant model. Remarkably, this single variable enabled the model to achieve a 56.9% correct classification rate in the

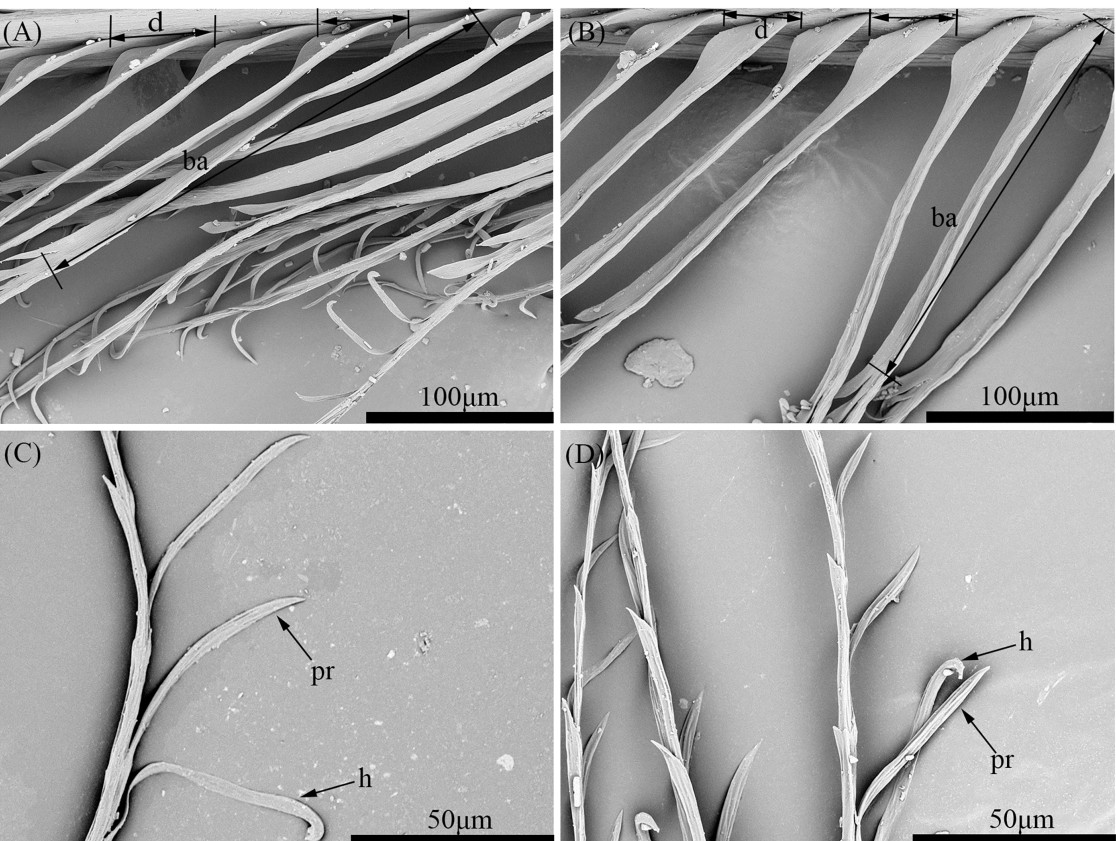

**Fig 1. Comparison of the distal barbules in the contour feathers from the Scaly-sided Merganser (A, C) and the Common Merganser (B, D).** *pr* prong, *d* inter-barbule distance, *ba* base, *h* hooklet.

**Table 2. Comparative microstructural characteristics of the contour feathers between Scaly-sided Merganser and Common Merganser.**

| Measurement | Scaly-sided Merganser | Common Merganser | U | P |
|---|---|---|---|---|
| Prong length (µm) | 50.0±5.64 | 49.1±4.91 | 1.712 | 0.087 |
| Base length (µm) | 266±15.1 | 261±12.7 | 1.849 | 0.064 |
| Inter-barbule distance (µm) | 47.2±2.33 | 47.0±2.22 | 0.166 | 0.869 |
| Hooklet number | 4.50±0.52 | 4.43±0.49 | 1.011 | 0.312 |
| Prong number | 6.51±0.63 | 6.37±0.60 | 1.499 | 0.134 |

\* Significant difference from the Common Merganser at P<0.05.

\*\* Significant difference from the Common Merganser at P<0.01.

**Table 3. Stepwise discriminant analyses to identify variables with the strongest predictive power to distinguish between two species in the contour feathers.**

| Predictor variable(s) | Wilks' Lambda statistic | Exact F statistic | P-value | Correct classification (%) |
|---|---|---|---|---|
| Base length | 0.966 | 5.544 | 0.02 | 56.9 |

leave-one-out cross-validation test for the selected specimens. This finding suggests that, even though contour feathers share overall microstructural similarities, the base length holds the potential to distinguish between the two species.

## Rectrices

A comparison of the inter-barbule distance of barbules from rectrices showed no significant difference (P=0.757) between the two species (Figs 2A and 2B, Table 4). However, significant disparities (P<0.01) were observed in the prong length, base length, hooklet number, and prong number of the rectrices between the two species (Fig 2, Table 4).

To pinpoint the most effective discriminative features for species differentiation, a stepwise discriminant analysis was applied to each variable of the rectrices. As presented in Table 5, the hooklet number (P<0.001) and prong length (P<0.001) were identified as significant discriminators, achieving an outstanding 91.3% correct classification rate in the leave-one-out cross-validation test among the selected specimens. These results suggest that these two features play a crucial role in distinguishing between the Scaly-sided Merganser and the Common Merganser.

## Down feathers

The downy barbules in both the Scaly-sided Merganser and the Common Merganser were structured with a base connected to the rachilla and a pennulum made up of cells that tapered distally, forming several expanded nodes. These nodes were slight protrusions taking on a triangular shape, and at the distal end of the barbules, the enlarged nodes were replaced by a few pairs of terminal prongs. Significant differences (P<0.01) were detected in the node number, distance between nodes, internode width, and barbule length of the down feathers between the two species (Fig 3, Table 6).

A stepwise discriminant analysis was then conducted on each variable of the down feathers. As detailed in Table 7, the barbule length (P<0.001), node number (P<0.001), and distance between nodes (P<0.001) were found to be highly effective discriminators. Employing the leave-one-out cross-validation test, these three features facilitated a perfect 100% correct classification rate among the selected specimens. These results suggest that these three characteristics can accurately distinguish Scaly – sided Merganser from Common Merganser.

## Discussion

Previous studies have demonstrated that feather microstructures exhibit variations not only among different species but also between sexes within the same species [4–8]. In this study, we identified significant differences in the prong length,

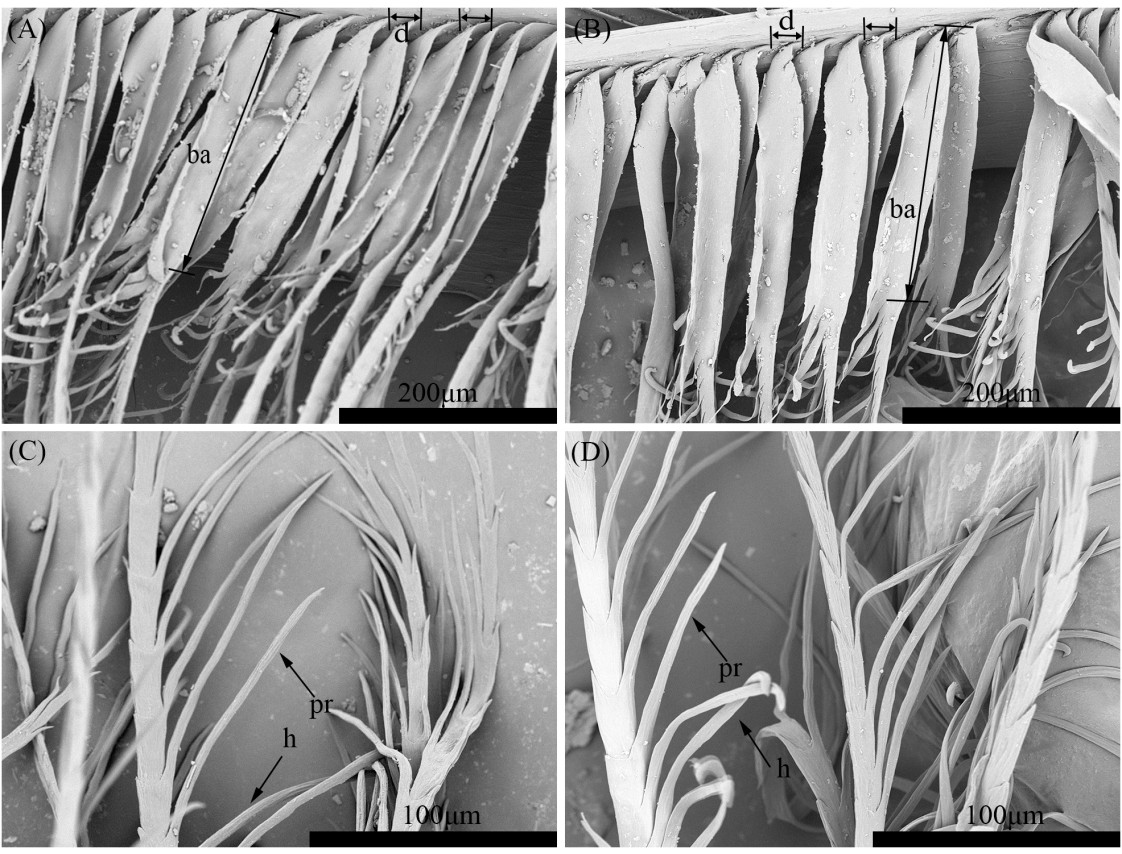

**Fig 2. Comparison of the distal barbules in the rectrices from the Scaly-sided Merganser (A, C) and the Common Merganser (B, D).** *pr* prong, *d* inter-barbule distance, *ba* base, *h* hooklet.

**Table 4. Comparative microstructural characteristics of rectrices between Scaly-sided Merganser and Common Merganser.**

| Measurement | Scaly-sided Merganser | Common Merganser | U | P |
|---|---|---|---|---|
| Prong length (µm) | 118±10.0** | 105±10.0 | 7.211 | 5.559e-13 |
| Base length (µm) | 276±13.6** | 248±16.9 | 8.456 | 2.756e-17 |
| Inter-barbule distance (µm) | 37.3±2.40 | 37.2±2.50 | 0.309 | 0.757 |
| Hooklet number | 8.44±0.50** | 6.46±0.82 | 10.340 | 4.643e-25 |
| Prong number | 12.6±1.01** | 11.1±1.03 | 8.462 | 2.629e-17 |

* Significant difference from the Common Merganser at P<0.05.

** Significant difference from the Common Merganser at P<0.01.

**Table 5. Stepwise discriminant analyses to identify variables with the strongest predictive power to distinguish between two species in the rectrices.**

| Predictor variable(s) | Wilks' Lambda statistic | Exact F statistic | P-value | Correct classification (%) |
|---|---|---|---|---|
| Hooklet number | 0.320 | 335.059 | 6.910e-41 | 91.3 |
| Prong length | 0.298 | 184.690 | 5.703e-42 | |

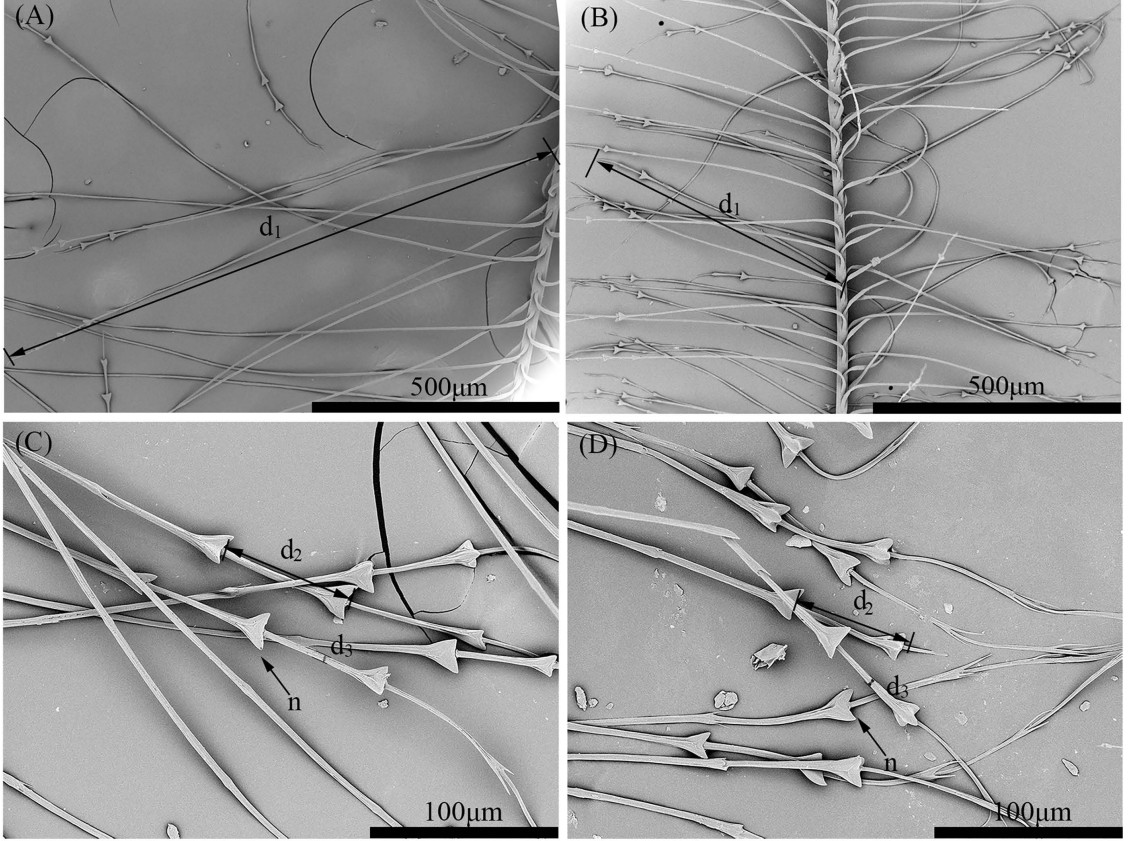

**Fig 3. Comparison of the downy barbules in the down feathers from the Scaly-sided Merganser (A) and the Common Merganser (B).** $n$ node, $d_1$ barbule length, $d_2$ distance between nodes, $d_3$ internode width.

**Table 6. Comparative Microstructural Characteristics of down feathers between Scaly-sided Merganser and Common Merganser.**

| Measurement | Scaly-sided Merganser | Common Merganser | $U$ | $P$ |
|---|---|---|---|---|
| Node number | 2.51±0.50** | 1.19±0.57 | 9.994 | 1.621e-23 |
| Distance between nodes (µm) | 69.5±4.36** | 58.8±2.85 | 10.297 | 7.304e-25 |
| Internode width (µm) | 4.18±0.51** | 3.80±0.38 | 4.584 | 5.000e-6 |
| Barbule length (µm) | 1329±165** | 586±33.3 | 10.921 | 9.104e-28 |

* Significant difference from the Common Merganser at $P<0.05$.

** Significant difference from the Common Merganser at $P<0.01$.

**Table 7. Stepwise discriminant analyses to identify variables with the strongest predictive power to distinguish between two species in down feathers.**

| Predictor variable(s) | Wilks' Lambda statistic | Exact F statistic | *P*-value | Correct classification (%) |
|---|---|---|---|---|
| Barbule length | 0.093 | 1538.162 | 2.448e-83 | 100 |
| Node number | 0.083 | 868.846 | 1.234e-85 | |
| Distance between nodes | 0.073 | 656.018 | 3.381e-88 | |

base length, hooklet number, and prong number of distal barbules in rectrices between the two species. Similarly, in down feathers, significant variations were observed in the node number, distance between nodes, internode width, and barbule length of downy barbules. These findings highlight the potential of these microstructural features as effective markers for differentiating between the Scaly-sided Merganser and the Common Merganser.

It is important to note that previous studies have reported differences in diagnostic feather traits between adults, juveniles, and fledglings [2,21–23]. Similarly, many studies have also pointed out that there is a certain degree of variation within species, between individuals, and even within the same feather [2,21–23]. Therefore, relying solely on differences in the length and distance of microstructural features of feathers, without considering the characteristics of the structure itself, may not always be reliable for species identification, especially when the selected feather types exhibit similar microstructural patterns. However, in this study, we randomly collected feather samples of the Scaly-sided Merganser after observing a group of them molting and leaving. As a result, we were unable to classify the Scaly-sided Merganser samples by age and instead analyzed all the samples together. Moreover, information on the age and exact number of individuals was unknown for most Common Merganser feather samples. However, we still found some microstructural characteristics that differed significantly between the two species. Based on these findings, we speculate that these differences were likely to stem from species-specific traits.

The microstructure of feathers provides an important reference for the identification of bird species [1]. Although species identification may be achieved with a single feather characteristic in many situations, it is strongly advisable to utilize multiple characteristics [6]. In situations such as ground investigations, criminal cases, or airplane collision incidents, conducting a thorough analysis of multiple microstructural features from the same feather is essential to determine its species of origin. Even if a single parameter (barbule length) shows a statistically significant difference and no apparent overlap between species (Table 6), identification based solely on this parameter may still be insufficient due to potential intraspecific variation and spatial differences in feather microstructure across different body regions. Stepwise discriminant analysis can address this limitation by selecting and combining the most discriminatory variables, thereby reducing the risk of misclassification. Based on the above considerations, in this study, stepwise discriminant analysis was also conducted on all microstructural characteristics of contour feathers, rectrices, and down feathers, with the aim of improving the accuracy and reliability of species identification. It is worth noting that only the barbule length, node number, and distance between nodes of the down feathers were used as predictor variables to effectively and accurately differentiate Scaly-sided Merganser from common Merganser. This discrepancy in taxonomic resolution across feather types presumably reflects contrasting functional constraints. Contour feathers and rectrices are strongly molded by ecological demands such as flight and display, predisposing them to convergent evolution [24]. Down feathers, specialized primarily for thermoregulation, experience relaxed selective pressure, retain phylogenetically conserved microstructures, and preserve greater species-specific information [25]. Therefore, the barbule length, node number, and distance between nodes of down feathers may be considered crucial morphological markers for differentiating the Scaly-sided Merganser from the Common Merganser.

However, the study has certain limitations. First, the specific collection areas of down feathers were not recorded. Given that down feathers from different body parts of birds may exhibit distinct microstructural characteristics due to functional differences, the lack of this information not only introduces uncertainties in the results but may also undermine the study's reproducibility and comparability across different research works. Although the down feather samples of Scaly-sided Mergansers in this study were randomly collected, theoretically covering various body parts, and the analysis of down feathers from different parts of two Common Merganser individuals revealed minimal microstructural variations within the species but significant differences compared to Scaly-sided Mergansers, the absence of collection location information may still interfere with the interpretation of the results. Second, the age of the sample individuals is unknown. Since feather microstructure may change with growth stages, age distribution biases between species could introduce systematic errors and compromise the reliability of interspecies comparisons. Third, due to limitations in the experimental

materials (unclear number of individuals), intraspecific variation in the barbule characteristics of the Scaly-sided Merganser and Common Merganser has not been explored. In light of the above limitations, relying solely on microstructural features such as the barbule length, node number, and distance between nodes of down feathers is insufficient to support accurate species identification between the two species. Nevertheless, these metrics can still be regarded as potential morphological cues for distinguishing between the Scaly-sided Merganser and the Common Merganser, and their diagnostic value needs to be confirmed by expanding the sample size, standardizing sampling procedures, and conducting statistical validation.

To address these limitations, future research should adopt a multi-faceted approach. First, a systematic sampling strategy is crucial, involving the collection of feathers from specific anatomical regions to account for potential spatial variations in microstructure. Second, sample collection should be extended to include diverse age groups (juveniles, sub-adults, and adults) and both sexes across multiple geographic populations, enabling stratified analyses to disentangle the effects of age, sex, and geography on feather morphology. Third, integrating advanced molecular techniques, such as genomics and proteomics, with traditional morphological analyses can uncover the genetic and biochemical mechanisms underlying feather microstructure, thereby enhancing the accuracy of species identification. By implementing these comprehensive improvements, we can elevate the scientific rigor of feather microstructure studies, laying a more robust foundation for avian species identification and ecological research.

## Conclusion

This study provides the first analysis of the feather microstructures of the Scaly-sided Merganser and the Common Merganser. We identified significant differences in several feather microstructural characteristics between the two species. In rectrices, the prong length, base length, hooklet number, and prong number of distal barbules varied distinctly. Down feathers also showed significant variations in the node number, distance between nodes, internode width, and barbule length of downy barbules. By conducting a stepwise discriminant analysis with the application of the leave-one-out cross-validation test, we found that the barbule length, node number, and distance between nodes of down feathers could achieve 100% correct classification. Consequently, the barbule length, node number, and distance between nodes of down feathers, may serve as potentially useful morphological markers for differentiating the two species.

## Supporting information

**S1 File. Full sample data on the microstructural characteristics of feathers in both the Scaly-sided Merganser and the Common Merganser.** The first sheet ("Contour feathers") includes the complete data on the microstructural characteristics of the barbules for the contour feathers of both species. The second sheet ("Rectrices") provides the complete data on the microstructural characteristics of the barbules for the rectrices of both species. The third sheet ("Down feathers") contains the complete data on the microstructural characteristics of the barbules for the down feathers of both species.
(XLSX)

## Acknowledgments

We sincerely thank the volunteers for providing feather samples of the Common Merganser. We also thank AJE for assistance with the revision of the paper.

## Author contributions

**Conceptualization:** Donghong Li, Shiyu Zhang.

**Formal analysis:** Donghong Li, Shiyu Zhang, Yongbin Zhao.

**Funding acquisition:** Yongbin Zhao.

**Investigation:** Donghong Li.

**Methodology:** Shiyu Zhang.

**Resources:** Guodong Yi.

**Supervision:** Shiyu Zhang, Ran Tian, Yongbin Zhao.

**Visualization:** Donghong Li, Yongbin Zhao.

**Writing – original draft:** Donghong Li.

**Writing – review & editing:** Shiyu Zhang, Ran Tian, Yongbin Zhao, Guodong Yi.

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
