## [Decision Letter · Decision Letter 0]

4 Mar 2025

Dear Dr. Zhao,

Thank you for submitting your manuscript to PLOS ONE. After careful consideration, we feel that it has merit but does not fully meet PLOS ONE’s publication criteria as it currently stands. Therefore, we invite you to submit a revised version of the manuscript that addresses the points raised during the review process.

The reviewers raise several important concerns regarding the feasibility of finding diagnostic species-level differences in the current study. I especially echo Reviewer 2's concerns regarding the very low sample size for common merganser. Adding feathers from additional individuals to the analysis will greatly increase the strength of the authors' arguments, and may also serve to alleviate some of Reviewer 1's concerns regarding individual-level differences (age, feather type).

We look forward to receiving your revised manuscript.

Kind regards,

Alex Slavenko

Academic Editor

PLOS ONE

Journal Requirements:

“This work was financially supported by the Jilin Provincial Department of Science and Technology (grant number: 20220101331JC). We thank AJE for assistance with the revision of the paper.”

6. We note that your Data Availability Statement is currently as follows: All relevant data are within the manuscript and its Supporting Information files.

Additional Editor Comments:

The reviewers bring up some important considerations regarding the ability to infer diagnostic differences between the species based on the presented data. I especially echo Reviewer 2's concern regarding the low sample size for Common Merganser - feather samples from additional individuals will add considerable strength to the authors' claims, and may also alleviate some of Reviewer 1's concerns regarding individual-level differences (age, feather location).

Reviewers' comments:

Reviewer's Responses to Questions

**Comments to the Author**

1. Is the manuscript technically sound, and do the data support the conclusions?

Reviewer #1: Partly

Reviewer #2: No

2. Has the statistical analysis been performed appropriately and rigorously?

Reviewer #1: N/A

Reviewer #2: I Don't Know

3. Have the authors made all data underlying the findings in their manuscript fully available?

Reviewer #1: No

Reviewer #2: Yes

4. Is the manuscript presented in an intelligible fashion and written in standard English?

Reviewer #1: No

Reviewer #2: Yes

Reviewer #1: The authors stated that the microstructures present in the breast feathers, flank feathers, rectrices, and down feathers are similar. However, they also noted that "Notably, variations were observed in the length of the base handle from distal barbules in breast feathers." It is surprising to see how species identification could be scientifically determined solely based on the length and distance of feather microstructures, especially when all selected feather types exhibit similar microstructures. Given that measurements will vary according to age and feather location, the authors' claim may be invalid. Without the presence of a distinct microstructure, species identification between the two studied species would not be feasible.

Reviewer #2: Overview: It is clear that the authors have dedicated much time and effort towards outlining and distinguishing these two species’ morphological barbule characters. There do appear to be differences in quite a few of these characters. However, I would heavily caution this interpretation on the basis of only one individual (Common Merganser) compared to what appears to be approx. 20 Scaly-sided Merganser individuals. It’s unclear as to which of the characters the authors recommend could/should be used to accurately discern Scaly-sided from Common Merganser – or whether all of the characters for multiple feather types are required to be taken together to accurately distinguish these two species from one another. This would be helpful for a practical application of these data, as the authors talk about it being useful for. And finally, a suggestion that a methods figure would greatly improve the ability of another scientist to understand these measurements and to be able to replicate these measurements on SEM images as this author has.

Abstract: Can you add the take home of what individual morphological barbule characters are recommended to be used to distinguish these two species?

Line 45 – Add Laybourne after Roxie to indicate who you are referring to

Line 46- A notable example of what? Might add that this is a notable example of the usefulness of these characters/notable example of the practical application of these characters etc.

Line 46 – Change ‘common starling’ to ‘European Starling’ (capitalization and common English name change)

Line 53 – concern regarding – maybe replace this English phrasing to : interest in. I don’t think there are concerns regarding these topics, but rather people heavily invested in examining and detailing these aspects of Scaly-sided Merganser biology.

Line 56 – ‘to address these issues – similar adjustment as above. Something like ‘answer questions related to these topics.’ Instead, is a better overview of the goal.

Line 57- direct capture OF blood. I think you’re referring to the direct capture of birds in order to take samples from them in which case, replace the word ‘of’ with ‘in order to take’ and that should more accurately describe what is difficult.

Line 60- classified as a (not the) species

Line 64 - …as ‘molted/found/undetermined’ feather samples collected. Typically feather samples don’t come from both species, but feather samples collected from an undetermined locality or found feather samples could come from both species or molted feathers found/taken from the ground are all feather samples that could come from both species.

Line 71 – morphological differentiation, in theory, wouldn’t be imperative but it does become an incredibly useful tool before sequencing DNA. I’d just change that particular word (imperative).

Materials/Methods

What was the condition of the feathers examined? Were more degraded feathers more difficult but still possible or impossible to see barbule characters?

Line 83 – How were approx. 20 individuals determined for Scaly-sided Mergansers? Was this genetically verified as 20 individuals? By a repeat number of parts (for example there were at least 20 right outermost primaries found/sampled? Sample size for each of the feather types per species.

Line 84 – Referred as ‘classified according to Featherbase’. What was being classified and how was Featherbase used to classify them as such. I think just a general clarification of what they were classified as, using Featherbase will help readers understood how it was used.

Am I understanding this correctly that there was only ONE Common Merganser sampled and approx. 20 Scaly-sided Mergansers sampled?

Line 88 – Why was the middle part of the feather vane sampled when most barbule characters are more well defined at the base of the feather? Were pennaceous barbs used? The plumulaceous feathers at the base of the feather vane? Distal, intermediate basal or umbilical plumulaceous regions of the feather are typically used to describe which feather barbs were examined.

Line 172 – when ‘slender morphology’ is used – what is being meant? I think you might want to use more descriptors to more accurately describe the morphology.

Table 1 – The measurements taken, is there any inter-observer differences in measurements? How were measurements taken? Was there a program used to do the measuring?

These measurements aren’t super intuitive, and the terminology used differs slightly from Chandler and Carla Dove’s previous and clearly defined characters. For example, the “Length of base nodular” – is this referring to the length of the most basal node or is it referring to the length of the basal cell (before the basal node). So, to make it simple I think making an example figure of a feather / barbule etc and showing the different measurements taken would be incredibly useful to visualize each and allow for someone else to repeat the study/take the same measurements in the same fashion.

Line 210 – common mergers should be “Common Mergansers”

Table 2 & disc. Function – All of the variables listed in Table 2 are in the discriminant analysis which then can determine species ID at 100%, however, this then requires 3 different known feather types from an unknown individual bird in order to determine species? This is extremely unlikely in the pratical case of a feather found on the ground, in a criminal investigation or in an airplane strike. Does this disc. analysis work per feather type or what is the minimum number of features needed to accurately identify/categorize these two species?

Line 262 – multiple features – which combination of features then are recommended to make this identification if the individual characters found in Table 1 (with significant differences between species) cannot be used.

Line 271-272 – I’m not sure that this study on morphological barbule characters aids in the exploration of ecological characteristics, habitat, dietary preferences and breeding behaviors.. I think that may be a bit of a stretch?

**Do you want your identity to be public for this peer review?** For information about this choice, including consent withdrawal, please see our Privacy Policy

Reviewer #1: No

Reviewer #2: No

---

## [Author Response · Author response to Decision Letter 1]

2 Apr 2025

Response to Editor

We are truly grateful to the reviewer for your comments. We deeply understand your legitimate concerns about our initial interpretation that was based on only one Common Merganser individual. Clearly, relying on a single sample was a precarious approach. The Common Merganser is a protected species, which makes it extremely challenging to gather a substantial number of samples. To address this issue, we obtained an adult male Common Merganser carcass from a volunteer in Shenyang, Liaoning Province, China. We also received multiple Common Merganser feathers from a volunteer in Fuzhou, Fujian Province, China.

Response to Reviewer #1

We thank the reviewer for their comments and for raising an important concern regarding the species identification based on microstructural differences in feathers. We fully acknowledge that relying merely on the length and distance variations of feather microstructural features, without considering the overall structural characteristics, may not always yield reliable results for species identification—especially when different feather types exhibit similar microstructural patterns.

In this study, we randomly collected feather samples of the Scaly-sided Merganser after observing a group of them molting and leaving. As a result, we were unable to classify the Scaly-sided Merganser samples by age and instead analyzed all the samples together. Specifically, 20 feathers of different types were randomly selected, including flank contour feathers, rectrices, and down feathers. In addition, we further expanded the sample size of Common Merganser feathers. These feathers were procured from diverse sources: an adult female Common Merganser specimen preserved at Jilin Normal University in Siping, Jilin Province, China; an adult male Common Merganser carcass donated by a volunteer from Shenyang, Liaoning Province, China; and a large quantity of feathers supplied by volunteers in Fuzhou, Fujian Province, China, with information on the age and exact number of individuals unknown for Common Merganser feather samples. Finally, three feathers per type were collected from each individual and randomly selected 14 feathers of each type from the Common Merganser samples provided by Fuzhou volunteer. These samples constituted the basis for analysis.

However, we still found some microstructural characteristics that differed significantly between the two species. Based on these findings, we speculate that these differences were likely to stem from species-specific traits rather than age-related factors. Furthermore, stepwise discriminant analysis further validated the discriminatory power of all microstructural characteristics. For contour feathers, incorporating the base length into the model achieved a 56.9% correct classification rate. In rectrices, the hooklet number and prong length emerged as key discriminators, with a correct classification rate of 91.3%. Most notably, the barbule length, node number, and inter-node distance of down feathers demonstrated exceptional discriminative capabilities, attaining a perfect 100% correct classification rate. Consequently, the barbule length, node number, and inter-node distance of down feathers, may be considered as crucial morphological markers for differentiating the Scaly-sided Merganser from the Common Merganser.

Species identification using the microstructural characteristics of feathers offers several advantages, including accelerating laboratory species identification, reducing costs, and enhancing work efficiency. To further advance research in this field, achieving accurate identification of feathers of unknown origin is a key goal. Future studies should consider expanding the sample scope to include more individuals of varying ages and sexes, thereby strengthening the generalizability of the research findings. Moreover, integrating multidisciplinary techniques such as genomics and bioinformatics to explore the genetic underpinnings of feather microstructural characteristics will provide a more solid theoretical foundation for species identification.

Thank you again for your valuable input.

Response to Reviewer #2

We sincerely appreciate your comprehensive review and constructive insights.

First, we fully understand your concern regarding the interpretation based on only one individual (Common Merganser) initially. It is indeed a risky approach. Given that the Common Merganser is also a protected species, acquiring a large number of samples poses significant challenges. To address this issue, we obtained an adult male Common Merganser carcass from a volunteer in Shenyang, Liaoning Province, China. We also received multiple Common Merganser feathers from a volunteer in Fuzhou, Fujian Province, China, with information on the age and exact number of individuals unknown for Common Merganser feather samples. Additionally, the exact number of individual Scaly-sided Mergansers involved in this study cannot be confirmed. Regarding the collection of Scaly-sided Merganser feathers from Longhu Mountain in Jiangxi Province, China, we observed a group of over 20 Scaly-sided Mergansers through binoculars during the moulting period. To minimize any disturbance, we randomly collected feathers only after the birds had left the site. Therefore, we were unable to determine the exact number of individuals, nor could we classify the samples based on age. Given these limitations, we combined all the collected samples for analysis and randomly selected 20 samples of different feather types, including contour feathers, rectrices, and down feathers.

Secondly, several microstructural characteristics of feathers exhibited significant variations between the two species. In rectrices, significant variations were observed in the prong length, base length, hooklet number, and prong number of distal barbules. Similarly, down feathers exhibited marked differences in the node number, inter-node distance, internode width, and barbule length of downy barbules. Stepwise discriminant analysis further validated the discriminatory power of all microstructural characteristics. For contour feathers, incorporating the base length into the model achieved a 56.9% correct classification rate. In rectrices, the hooklet number and prong length emerged as key discriminators, with a correct classification rate of 91.3%. Most notably, the barbule length, node number, and inter-node distance of down feathers demonstrated exceptional discriminative capabilities, attaining a perfect 100% correct classification rate. Consequently, the barbule length, node number, and inter-node distance of down feathers, may be considered as crucial morphological markers for differentiating the Scaly-sided Merganser from the Common Merganser.

Finally, to improve the clarity of our methods, we have added Table 1, which clearly describes all measurement methods and parameters. We have also updated Figs. 1, 2, and 3 with corresponding labels to better illustrate the measurement process, aiming to facilitate other scientists in understanding and replicating our measurements on SEM images.

Thank you again for your thorough review and constructive feedback.

1 Abstract: Can you add the take home of what individual morphological barbule characters are recommended to be used to distinguish these two species?

Additions have been made. “all the microstructural features of down feathers, particularly barbule length, may be considered as crucial morphological markers for differentiating the two species.”

2 Line 45 – Add Laybourne after Roxie to indicate who you are referring to

Line 46- A notable example of what? Might add that this is a notable example of the usefulness of these characters/notable example of the practical application of these characters etc.

Line 46 – Change ‘common starling’ to ‘European Starling’ (capitalization and common English name change.

Additions and modifications have been made. “A notable example of the practical application of these characters is Roxie Laybourne's work, where she identified the European Starling (Sturnus vulgaris) as the cause of an aviation accident by analyzing feathers found in the wreckage[3].”

3 Line 53 – concern regarding – maybe replace this English phrasing to: interest in. I don’t think there are concerns regarding these topics, but rather people heavily invested in examining and detailing these aspects of Scaly-sided Merganser biology.

Modifications have been made. “Consequently, there is substantial interest in various aspects of Scaly-sided Merganser, including population size [10–12], habitat selection [13,14], genetic diversity [15], migration connections [16], and breeding ecology [17].”

4 Line 56 – ‘to address these issues – similar adjustment as above. Something like ‘answer questions related to these topics.’ Instead, is a better overview of the goal.

Modifications have been made. “In addition to field observations, molecular biology and isotope tracing are often used to answer questions related to these topics.”

5 Line 57- direct capture OF blood. I think you’re referring to the direct capture of birds in order to take samples from them in which case, replace the word ‘of’ with ‘in order to take’ and that should more accurately describe what is difficult.

Modifications have been made. “However, due to the rarity and high vigilance of Scaly-sided Mergansers, direct capture in order to take blood or tissue samples is becoming increasingly challenging.”

6 Line 60- classified as a (not the) species

Modifications have been made. “It is noteworthy that Common Merganser (M. merganser), classified as a species of least concern according to the IUCN Red List, is distinguished as the largest and most widely distributed species within the genus Mergus in China[18].”

7 Line 64 - …as ‘molted/found/undetermined’ feather samples collected. Typically feather samples don’t come from both species, but feather samples collected from an undetermined locality or found feather samples could come from both species or molted feathers found/taken from the ground are all feather samples that could come from both species.

Modifications have been made. “This presents a challenge for ecologists, as feather samples, whether molted, found, or from an undetermined locality, may originate from both species.”

8 Line 71 – morphological differentiation, in theory, wouldn’t be imperative but it does become an incredibly useful tool before sequencing DNA. I’d just change that particular word (imperative).

Modifications have been made. “Therefore, morphological differentiation of feather samples is an incredibly useful tool before sequencing DNA.”

9 Materials/Methods: What was the condition of the feathers examined? Were more degraded feathers more difficult but still possible or impossible to see barbule characters?

Since the birds were observed moulting with binoculars and collected immediately when they left, the feathers examined were in good condition and not degraded. This has been written under Materials/Methods.

10 Line 83 – How were approx. 20 individuals determined for Scaly-sided Mergansers? Was this genetically verified as 20 individuals? By a repeat number of parts (for example there were at least 20 right outermost primaries found/sampled? Sample size for each of the feather types per species.

A group of more than 20 Scaly-sided Mergansers was observed through binoculars during the moulting period. However, to minimize disturbance, feathers were only collected after the birds had left the site. As a result, the exact number of individual Scaly-sided Mergansers involved in this study cannot be confirmed. From the collected feathers, 20 feathers of different types were randomly selected, including flank contour feathers, rectrices, and down feathers.

Additionally, feathers of Common Merganser were obtained from the following sources: an adult female Common Merganser specimen from the collection of Jilin Normal University in Siping City, Jilin Province, China; an adult male Common Merganser carcass provided by a volunteer from Shenyang City, Liaoning Province, China; and multiple feathers of the Common Merganser provided by a volunteer from Fuzhou City, Fujian Province, China. Finally, three feathers of each type were collected from each individual. Moreover, 14 feathers of each type were randomly chosen from the Common Merganser feather samples provided by volunteers.

Hence, the sample size was 20 per species per feather type.

11 Line 84 – Referred as ‘classified according to Featherbase’. What was being classified and how was Featherbase used to classify them as such. I think just a general clarification of what they were classified as, using Featherbase will help readers understood how it was used. Am I understanding this correctly that there was only ONE Common Merganser sampled and approx. 20 Scaly-sided Mergansers sampled?

Yes, you are correct. We sampled one Common Merganser and approximately 20 Scaly - sided Mergansers.

There is a feather specimen of an adult Scaly-sided Merganser in Featherbase. We performed a simple classification of the randomly collected feathers based on Featherbase. However, after further consideration, we revised the paper, as distinguishing the contour feathers may not be accurately done solely based on Featherbase. Therefore, we removed the analysis of the breast feathers. Since the Scaly-sided Merganser is known for its fish-scale-like flank feathers, which can be easily identified from feather samples, we retained the observation and analysis of the contour feathers of the flank.

12 Line 88 – Why was the middle part of the feather vane sampled when most barbule characters are more well defined at the base of the feather? Were pennaceous barbs used? The plumulaceous feathers at the base of the feather vane? Distal, intermediate, basal or umbilical plumulaceous regions of the feather are typically used to describe which feather barbs were examined.

The terminology used to describe the feather sections was inaccurate and has been revised. Additionally, adjustments have been made to the Materials and Methods section.

13 Line 172 – when ‘slender morphology’ is used – what is being meant? I think you might want to use more descriptors to more accurately describe the morphology.

The down feathers section has been revised and adjusted.

14 Table 1 – The measurements taken, is there any inter-observer differences in measurements? How were measurements taken? Was there a program used to do the measuring? These measurements aren’t super intuitive, and the terminology used differs slightly from Chandler and Carla Dove’s previous and clearly defined characters. For example, the “Length of base nodular” – is this referring to the length of the most basal node or is it referring to the length of the basal cell (before the basal node). So, to make it simple I think making an example figure of a feather / barbule etc and showing the different measurements taken would be incredibly useful to visualize each and allow for someone else to repeat the study/take the same measurements in the same fashion.

Indeed, the terminology used was somewhat inaccurate, and these measurements aren't very intuitive. We have revised the terminology throughout the paper. In Table 1, all the measurement methods and parameters are clearly explained, and corresponding labels are provided in Figs 1, 2 and 3 to better illustrate the measurement process.

15 Line 210 – common mergers should be “Common Mergansers”

Modifications have been made.

16 Table 2 & disc. Function – All of the variables listed in Table 2 are in the discriminant analysis which then can determine species ID at 100%, however, this then requires 3 different known feather types from an unknown individual bird in order to determine species? This is extremely unlikely in the pratical case of a feather found on the ground, in a criminal investigation or in an airplane strike. Does this disc. analysis work per feather type or what is the minimum number of features needed to accurately identify/categorize these two species?

Indeed, obtaining multiple different known feather types (such as feathers found on the ground, in criminal investigations, or in an airplane strike) is extremely unlikely in practical cases. In

---

## [Decision Letter · Decision Letter 1]

30 Apr 2025

Dear Dr. Zhao,

Thank you for submitting your manuscript to PLOS ONE. After careful consideration, we feel that it has merit but does not fully meet PLOS ONE’s publication criteria as it currently stands. Therefore, we invite you to submit a revised version of the manuscript that addresses the points raised during the review process.

Please carefully address the comments by the new reviewer, particularly those pertaining to cross-validation of the classification model and possible caveats around anatomical regions of feathers.

We look forward to receiving your revised manuscript.

Kind regards,

Alex Slavenko

Academic Editor

PLOS ONE

Additional Editor Comments :

I commend the authors for responding carefully and meticulously to the comments from the first round of revision. The addition of new samples is very welcome.

The original reviewers of the manuscript were, unfortunately, unavailable to review the revised version. However, I have received an additional review from a new reviewer. The new reviewer has identified some potential issues with the study that I think should be addressed in another round of revisions.

Reviewers' comments:

Reviewer's Responses to Questions

**Comments to the Author**

Reviewer #3: (No Response)

2. Is the manuscript technically sound, and do the data support the conclusions?

Reviewer #3: No

3. Has the statistical analysis been performed appropriately and rigorously?

Reviewer #3: (No Response)

4. Have the authors made all data underlying the findings in their manuscript fully available?

Reviewer #3: (No Response)

5. Is the manuscript presented in an intelligible fashion and written in standard English?

Reviewer #3: Yes

Reviewer #3: 1. The authors mention that flank contour feathers were included in the analysis; however, it is unclear how exactly the flank contour feathers were identified among the feathers provided by volunteers. This is particularly important for the Common Merganser (Mergus merganser), in which males have a large portion of the underside (including the breast and belly) covered with white plumage, similar in appearance to the flanks. This raises the risk of misidentifying the feather type during sampling, especially if the feathers were collected without access to the carcass. I recommend clarifying which morphological or anatomical criteria were used to identify flank feathers specifically, and how the possibility of including feathers from other similarly colored body areas (e.g., breast or belly) was excluded.

2. It is unclear in what form the feathers were provided by the volunteers. If they were submitted as a general batch without separation by individual bird or by anatomical origin, there is a considerable risk that feathers not corresponding to the intended body region were included in the analysis, or that multiple feathers from the same individual were analyzed. This situation could distort within-group variability and lead to an overestimation of interspecific differences. A similar issue arises with the Scaly-sided Merganser samples, as the exact number of contributing individuals is also unknown. In this context, the reported classification accuracy—especially the 100% result for down feathers—raises concerns. While interspecific differences may indeed be present, I believe it is essential to perform additional validation of the model, for example, using cross-validation techniques, to rule out overfitting and confirm the robustness of the results.

3. The manuscript does not specify the anatomical region from which the down feathers were collected—despite the fact that these feathers yielded the highest classification accuracy (100%). Given that the microstructure of down feathers may vary depending on their position on the bird's body, the lack of this information complicates both the interpretation of the results and the reproducibility of the study. It is possible that the morphological features of downy barbules are consistent across body regions; however, before making generalized conclusions, it would be appropriate to investigate this question in more detail—or at the very least, to acknowledge it as a potential limitation of the method.

4. The authors state that they were unable to determine the age of the individuals from which the feathers were collected, and therefore performed the analysis without separating samples by age group. However, this approach leaves open the possibility of unequal age distribution between species. For example, the Common Merganser sample might include up to five juvenile individuals, while the Scaly-sided Merganser sample may include only one. Given that microstructural characteristics of feathers—including barbule morphology, node count, and hooklet number—can vary with age, such asymmetry could influence the results of interspecific comparisons. It is recommended that this factor be at least addressed as a potential limitation of the study, particularly in the context of interpreting the high accuracy reported in species-level classification.

5. It would be helpful for international readers if the total number of feathers used in the study were stated more clearly, along with the number of each feather type analyzed per species. Currently, the relevant information is presented in a fragmented way across the text, which makes it difficult to reconstruct the structure of the dataset. Providing this information explicitly—either in a separate paragraph or in a table—would improve the clarity of the methodology and increase confidence in the results.

Positive Aspects of the Study

1. The authors analyzed the microstructure of three functionally distinct feather types—contour feathers, rectrices, and down feathers. This broadens the scope of the study and enhances the generalizability of its conclusions. Such an approach is particularly valuable when working with incomplete or randomly sourced feather samples, as it allows for an assessment of how diagnostic traits are expressed across different feather zones.

2. Barbs were sampled from both the left and right vanes of each feather for microstructural analysis. This is an important methodological choice, as it helps minimize bias due to possible asymmetry within a single feather and increases the reliability of the measurements obtained.

3. The study has strong practical implications. The potential to identify bird species based on feather microstructure is highly relevant in fields such as forensic science, aviation safety, and zoological diagnostics. The authors’ attempt to define morphological markers that could enable species identification even from isolated feather types is a valuable step forward, despite certain limitations in study design.

**Do you want your identity to be public for this peer review?** For information about this choice, including consent withdrawal, please see our Privacy Policy

Reviewer #3: **Yes:** Fayfer Victoria Valentinovna

---

## [Author Response · Author response to Decision Letter 2]

13 May 2025

Response to Reviewers

1.The authors mention that flank contour feathers were included in the analysis; however, it is unclear how exactly the flank contour feathers were identified among the feathers provided by volunteers. This is particularly important for the Common Merganser (Mergus merganser), in which males have a large portion of the underside (including the breast and belly) covered with white plumage, similar in appearance to the flanks. This raises the risk of misidentifying the feather type during sampling, especially if the feathers were collected without access to the carcass. I recommend clarifying which morphological or anatomical criteria were used to identify flank feathers specifically, and how the possibility of including feathers from other similarly colored body areas (e.g., breast or belly) was excluded.

We sincerely appreciate you for your insightful comments on the identification of flank contour feathers in our study. We acknowledge the challenge of distinguishing these feathers from other similarly coloured ones, especially in male common mergansers, which have white feathers on their breast, belly, and flanks. It should be emphasised that the feathers provided by a volunteer in Fuzhou were provided on the basis of anatomical position. On this basis, we checked these feathers against feathers in the herbarium. By observing the macrostructure of the feathers under the light microscope, we focused on morphological features such as the length, width, and curvature of the feather shaft. Flank contour feathers were found to be longer than those from the breast and belly, with a thicker and more curved shaft. These features differed significantly from feathers of other body parts, enabling us to effectively exclude feathers that might have been derived from other regions.

2� It is unclear in what form the feathers were provided by the volunteers. If they were submitted as a general batch without separation by individual bird or by anatomical origin, there is a considerable risk that feathers not corresponding to the intended body region were included in the analysis, or that multiple feathers from the same individual were analyzed. This situation could distort within-group variability and lead to an overestimation of interspecific differences. A similar issue arises with the Scaly-sided Merganser samples, as the exact number of contributing individuals is also unknown. In this context, the reported classification accuracy—especially the 100% result for down feathers—raises concerns. While interspecific differences may indeed be present, I believe it is essential to perform additional validation of the model, for example, using cross-validation techniques, to rule out overfitting and confirm the robustness of the results.

We apologize for the lack of clarity regarding the form in which feathers were provided by volunteers. Feathers donated by a volunteer in Fuzhou were collected based on anatomical position, and we have now explicitly stated this in the manuscript. Furthermore, all reported classification accuracies, including the 100% result for down feathers, were obtained using a leave-one-out cross-validation approach, a detail we have also added throughout the manuscript. These additions should address concerns about the robustness of our findings and potential overfitting of the model.

3�The manuscript does not specify the anatomical region from which the down feathers were collected—despite the fact that these feathers yielded the highest classification accuracy (100%). Given that the microstructure of down feathers may vary depending on their position on the bird's body, the lack of this information complicates both the interpretation of the results and the reproducibility of the study. It is possible that the morphological features of downy barbules are consistent across body regions; however, before making generalized conclusions, it would be appropriate to investigate this question in more detail—or at the very least, to acknowledge it as a potential limitation of the method.

4. The authors state that they were unable to d etermine the age of the individuals from which the feathers were collected, and therefore performed the analysis without separating samples by age group. However, this approach leaves open the possibility of unequal age distribution between species. For example, the Common Merganser sample might include up to five juvenile individuals, while the Scaly-sided Merganser sample may include only one. Given that microstructural characteristics of feathers—including barbule morphology, node count, and hooklet number—can vary with age, such asymmetry could influence the results of interspecific comparisons. It is recommended that this factor be at least addressed as a potential limitation of the study, particularly in the context of interpreting the high accuracy reported in species-level classification.

Thank you for your insightful feedback. We fully recognize the limitations of our study, primarily stemming from the lack of information regarding the collection sites of down feathers and the ages of sampled individuals. These gaps in data have the potential to influence feather microstructural characteristics, thereby affecting the generalizability and reliability of our results.

In response, we have added a detailed discussion of these limitations to the manuscript. Specifically, we note two key constraints. First, the specific collection areas of down feathers were not recorded. Given that down feathers from different body parts of birds may exhibit distinct microstructural characteristics due to functional differences, the lack of this information not only introduces uncertainties in the results but may also undermine the study's reproducibility and comparability across different research works. Although the down feather samples of Scaly-sided Mergansers in this study were randomly collected, theoretically covering various body parts, and the analysis of down feathers from different parts of two Common Merganser individuals revealed minimal microstructural variations within the species but significant differences compared to Scaly-sided Mergansers, the absence of collection location information may still interfere with the interpretation of the results. Second, the age of the sample individuals is unknown. Since feather microstructure may change with growth stages, age distribution biases between species could introduce systematic errors and compromise the reliability of interspecies comparisons.

To address these limitations, we propose that future research adopt a more comprehensive sampling strategy. This approach should include collecting feathers from defined anatomical regions (such as the chest, abdomen, and wings) and across diverse age groups (juveniles, sub-adults, and adults). By implementing these improvements, we aim to enhance the rigor of feather microstructure studies. We hope that our current findings, despite their limitations, can serve as a foundation for more robust investigations in this field.

5�It would be helpful for international readers if the total number of feathers used in the study were stated more clearly, along with the number of each feather type analyzed per species. Currently, the relevant information is presented in a fragmented way across the text, which makes it difficult to reconstruct the structure of the dataset. Providing this information explicitly—either in a separate paragraph or in a table—would improve the clarity of the methodology and increase confidence in the results.

We have addressed your feedback by introducing a dedicated section in the Methods section to explicitly report the total number of feathers (120 samples: 60 per species) and the distribution of feather types (20 samples each for flank contour feathers, rectrices, and down feathers per species). The sampling strategies for each species—including individual-specific collections and volunteer-donated samples—and sample preparation protocols are now clearly outlined in a structured paragraph, eliminating the previous fragmented presentation. This revision ensures that readers can readily reconstruct the dataset composition and methodology, enhancing the study’s clarity and replicability. By specifying the number of samples per feather type and species, as well as detailing the sampling and processing steps, we aim to provide a transparent foundation for our results and build confidence in the analytical framework. Thank you for your insightful suggestion, which has significantly improved the methodological description.

---

## [Decision Letter · Decision Letter 2]

15 Aug 2025

Dear Dr. Zhao,

Thank you for submitting your manuscript to PLOS ONE. After careful consideration, we feel that it has merit but does not fully meet PLOS ONE’s publication criteria as it currently stands. Therefore, we invite you to submit a revised version of the manuscript that addresses the points raised during the review process.

We look forward to receiving your revised manuscript.

Kind regards,

Alex Slavenko

Academic Editor

PLOS ONE

Journal Requirements:

Additional Editor Comments:

I have now received a review of your revised manuscript, conducted by one of the original reviewers from the original submission. While the reviewer overall thinks there is merit in this work, they have also identified a key issue which I agree absolutely must be resolved - the limitations of the study must be made clear, and the language tempered in some sections. The lack of detail about the exact sample sizes also ties in with the lack of detail about collection localities and personnel, about which we have raised a query with you via emails from the editorial office. You have responded to the email with the following answers:

Q1. Did you require a permit to collect the naturally shed feathers of the Scaly-Sided Merganser, an endangered species?

A1. The naturally shed feathers analyzed in this study were collected from publicly accessible areas outside the core protected zones of the species' wintering range. In these regions, non-invasive scientific activities, including the collection of naturally shed biological materials, are explicitly permitted under existing wildlife conservation laws without the need for special permits. The sampling process involved only the retrieval of feathers that had naturally molted and fallen to the ground, with no direct interaction, capture, or disturbance of live individuals or their ecological habitats.

Q2. Were all observations and sample collections performed on public and unprotected land? Why were field permits not required?

A2. All observations and sample collections were conducted in public and unprotected areas. In these regions, non-invasive scientific activities, including the collection of naturally shed biological materials, are explicitly permitted under existing wildlife conservation laws without the need for special permits. This is because the activities do not involve direct interaction, capture, or disturbance of live individuals or their habitats.

Q3. Could you please provide additional information about the expertise of the volunteers who provided your additional samples, particularly their institutional affiliations if applicable and any relevant qualifications they hold relating to bird species identification?

A3. The volunteer providing Common Merganser (Mergus merganser) feather samples is affiliated with Fuzhou Senyi Ecological Environment Engineering Co., Ltd. (hereinafter referred to as "the Company"), a professionally registered entity specializing in wildlife specimen restoration and conservation. The company is authorized by local forestry departments to handle non-protected wildlife specimens and has collaborated with national zoos on educational exhibitions. Its team includes certified taxidermists, senior engineers in ecology, and professionals trained in avian anatomy, ensuring compliance with scientific collection standards.

Q4. How did the volunteer collect the feather samples, and how did they determine the anatomical position of the feathers? You mention that they did not report the number of individuals from which the feathers were obtained.

A4. The feathers were collected from the corresponding anatomical positions on the preserved specimens. However, due to the nature of the collection process, the feathers were collected in bulk and subsequently mixed together during packaging and transportation. As a result, it was not possible to determine the exact number of individuals from whom the feathers were obtained.

I must stress that this information, as well as clarification about sample sizes and limitations as requested by the reviewer, absolutely must be included in the manuscript for it to adhere to principles of transparency and for it to be publishable. At this stage, I am afraid I must request another revision to address this.

Reviewers' comments:

Reviewer's Responses to Questions

**Comments to the Author**

Reviewer #2: (No Response)

2. Is the manuscript technically sound, and do the data support the conclusions?

Reviewer #2: No

3. Has the statistical analysis been performed appropriately and rigorously?

Reviewer #2: (No Response)

4. Have the authors made all data underlying the findings in their manuscript fully available?

Reviewer #2: (No Response)

5. Is the manuscript presented in an intelligible fashion and written in standard English?

Reviewer #2: Yes

Reviewer #2: Overall: The limitations of this study need to be further expanded on to make it clear what the practical application (forensic application as suggested by the authors intro) of using these measurements to identify unknown feathers is and sample size needs to be made more clear for each species and each feather type as also suggested in a previous reviewer comment.

As it stands, it is phrased in such a way that one could mistake these characters and measurements are clear and robust enough to differentiate the two species, reliably. The sample size is not robust enough, nor is the intraspecific variation in barbule characters described well enough to utilize these microscopic characters alone characters to identify these two species in a forensic context.

The clarification/isolation and identification of their different feather types (contour, rectrices and down feathers) would be extremely useful in helping parse apart any differences between species and within species variation. The practical applications are numerous and very clearly and well stated. I think if the limitations are more clearly stated, this is a useful addition to the body of literature out there on microscopic identification of feathers.

While stated mostly in the discussion a little expansion on these limitations may be helpful. We need to know individual variation in these characters (barbule length, how it relates to the number of nodes, does the distance between nodes vary along the length of a barbule, does the distance between nodes vary along where on the feather it is measured (basal barbules or more distal barbules)). Especially because of how overall less common this field of study is, it feels critical to outline these potential limitations for any new forensic scientist or solicitor/lawyer who may want to use these measurements to help identify an unknown feather – given the small sample size of individuals.

Additionally – a limitation on the forensic application of these data is important. When talking about stepwise discriminant analysis it is helpful to know what measurements are NEEDED to make a classification, and why a basic difference with statistically significant measurement of prong length isn’t sufficient to determine between the two species which do not have overlap in measurements (per table 2) and why the stepwise discriminant analysis is necessary – UNLESS we know that there is a lot of individual variation in those measurements and there actually is quite a bit of overlap along different areas of the feathers or different barbules from different areas of the bird.

Additional notes:

Abstract: “All the microstructural features of down feathers, particularly barbule length, may be considered as crucial morphological markers for differentiating the two species.” Whereas the morphological barbule characters described as capable of discerning Scaly-sided and Common Merganser are barbule length, node number, and inter-node distance of down feathers, …for differentiating the Scaly-sided Merganser from the Common Merganser. Update to include only the relevant morphological characters/take home characters suggested/recommended to distinguish these two species.

It is mentioned that approx. 20 individuals for Scaly-sided Mergansers were sampled and the authors have given context that there were at least 20 individuals in the flock observed and later collected feathers from. It’s important to understand that this does not mean that all 20 individuals were sampled and should not be phrased as such without genetic confirmation of the minimum number of individuals that these feathers could have originated from. The paper stating that 20 individuals of this species are represented here is stating the maximum number of individuals the feathers could originate from, which is very likely an overestimate.

For Common Merganser, it now appears that you have at least 3 individuals represented, but there is no way to claim that there were 20 Common Mergansers represented by the 2 known carcasses (ad. Female, ad male) and the multiple feathers from volunteer. The claim that this would represent at least 20 individuals is misleading and an overrepresentation. Please make explicit the sample size for each species (Common Merganser, min number of individuals represented = 3) or (Scaly-sided Merganser, min number of individuals represented = X?) and then you can also detail that pooled by species at least 20 feathers of each feather type for each species were examined for this study. This would help clarify how many feathers were examined and from the minimum number of birds those feathers originated. Very helpful information to understand the breadth of representation in possible feather variation captured by this sampling.

Line 232 – Common Merganser, the C in Common should be capitalized.

**Do you want your identity to be public for this peer review?** For information about this choice, including consent withdrawal, please see our Privacy Policy

Reviewer #2: No

---

## [Author Response · Author response to Decision Letter 3]

25 Sep 2025

Response to Reviewers

1. The lack of detail about the exact sample sizes also ties in with the lack of detail about collection localities and personnel, about which we have raised a query with you via emails from the editorial office.

The details have been supplemented in the manuscript.

1. The limitations of this study need to be further expanded on to make it clear what the practical application (forensic application as suggested by the authors intro) of using these measurements to identify unknown feathers is and sample size needs to be made more clear for each species and each feather type as also suggested in a previous reviewer comment.

The limitations have been further expanded, and the sample size for each species and every feather type is now explicitly stated.

2. As it stands, it is phrased in such a way that one could mistake these characters and measurements are clear and robust enough to differentiate the two species, reliably. The sample size is not robust enough, nor is the intraspecific variation in barbule characters described well enough to utilize these microscopic characters alone characters to identify these two species in a forensic context.

Thank you for your thorough review and constructive comments. We fully agree that the original wording could easily lead readers to believe that the current sample size and barbule characters already suffice for species identification. To avoid any misinterpretation, we have systematically revised the relevant sections and softened the language accordingly. The specific changes are as follows:

Lines 39-41: “Consequently, the barbule length, node number, and distance between nodes of down feathers, may serve as potentially useful morphological markers for differentiating the Scaly-sided Merganser from the Common Merganser.”

Lines 267-272: “In light of the above limitations, relying solely on microstructural features such as the barbule length, node number, and distance between nodes of down feathers is insufficient to support accurate species identification between the two species. Nevertheless, these metrics can still be regarded as potential morphological cues for distinguishing between the Scaly-sided Merganser and the Common Merganser,”

Lines 296-298: “Consequently, the barbule length, node number, and distance between nodes of down feathers, may serve as potentially useful morphological markers for distinguishing between the two species.”

3. The clarification/isolation and identification of their different feather types (contour, rectrices and down feathers) would be extremely useful in helping parse apart any differences between species and within species variation. The practical applications are numerous and very clearly and well stated. I think if the limitations are more clearly stated, this is a useful addition to the body of literature out there on microscopic identification of feathers. While stated mostly in the discussion a little expansion on these limitations may be helpful. We need to know individual variation in these characters (barbule length, how it relates to the number of nodes, does the distance between nodes vary along the length of a barbule, does the distance between nodes vary along where on the feather it is measured (basal barbules or more distal barbules)). Especially because of how overall less common this field of study is, it feels critical to outline these potential limitations for any new forensic scientist or solicitor/lawyer who may want to use these measurements to help identify an unknown feather – given the small sample size of individuals.

The limitations have now been supplemented as follows: “Third, due to limitations in the experimental materials (unclear number of individuals), intraspecific variation in the barbule characteristics of the Scaly-sided Merganser and Common Merganser has not been explored. In light of the above limitations, relying solely on microstructural features such as the barbule length, node number, and distance between nodes of down feathers is insufficient to support accurate species identification between the two species. Nevertheless, these metrics can still be regarded as potential morphological cues for distinguishing between the Scaly-sided Merganser and the Common Merganser, and their diagnostic value needs to be confirmed by expanding the sample size, standardizing sampling procedures, and conducting statistical validation.”

4. Additionally – a limitation on the forensic application of these data is important. When talking about stepwise discriminant analysis it is helpful to know what measurements are NEEDED to make a classification, and why a basic difference with statistically significant measurement of prong length isn’t sufficient to determine between the two species which do not have overlap in measurements (per table 2) and why the stepwise discriminant analysis is necessary – UNLESS we know that there is a lot of individual variation in those measurements and there actually is quite a bit of overlap along different areas of the feathers or different barbules from different areas of the bird.

We agree that clarifying the rationale for adopting stepwise discriminant analysis in this study is important. The relevant section has been revised as follows (Lines 232–246): "The microstructure of feathers provides an important reference for the identification of bird species [1]. Although species identification may be achieved with a single feather characteristic in many situations, it is strongly advisable to utilize multiple characteristics [6]. In situations such as ground investigations, criminal cases, or airplane collision incidents, conducting a thorough analysis of multiple microstructural features from the same feather is essential to determine its species of origin. Even if a single parameter (barbule length) shows a statistically significant difference and no apparent overlap between species (Table 6), identification based solely on this parameter may still be insufficient due to potential intraspecific variation and spatial differences in feather microstructure across different body regions. Stepwise discriminant analysis can address this limitation by selecting and combining the most discriminatory variables, thereby reducing the risk of misclassification. Based on the above considerations, in this study, stepwise discriminant analysis was also conducted on all microstructural characteristics of contour feathers, rectrices, and down feathers, with the aim of improving the accuracy and reliability of species identification.”

1. Abstract: “All the microstructural features of down feathers, particularly barbule length, may be considered as crucial morphological markers for differentiating the two species.” Whereas the morphological barbule characters described as capable of discerning Scaly-sided and Common Merganser are barbule length, node number, and inter-node distance of down feathers, …for differentiating the Scaly-sided Merganser from the Common Merganser. Update to include only the relevant morphological characters/take home characters suggested/recommended to distinguish these two species.

The abstract already states: “Consequently, the barbule length, node number, and distance between nodes of down feathers, may serve as potentially useful morphological markers for differentiating the Scaly-sided Merganser from the Common Merganser.” This sentence appears on Lines 39–41.

2. It is mentioned that approx. 20 individuals for Scaly-sided Mergansers were sampled and the authors have given context that there were at least 20 individuals in the flock observed and later collected feathers from. It’s important to understand that this does not mean that all 20 individuals were sampled and should not be phrased as such without genetic confirmation of the minimum number of individuals that these feathers could have originated from. The paper stating that 20 individuals of this species are represented here is stating the maximum number of individuals the feathers could originate from, which is very likely an overestimate. For Common Merganser, it now appears that you have at least 3 individuals represented, but there is no way to claim that there were 20 Common Mergansers represented by the 2 known carcasses (ad. Female, ad male) and the multiple feathers from volunteer. The claim that this would represent at least 20 individuals is misleading and an overrepresentation. Please make explicit the sample size for each species (Common Merganser, min number of individuals represented = 3) or (Scaly-sided Merganser, min number of individuals represented = X?) and then you can also detail that pooled by species at least 20 feathers of each feather type for each species were examined for this study. This would help clarify how many feathers were examined and from the minimum number of birds those feathers originated. Very helpful information to understand the breadth of representation in possible feather variation captured by this sampling.

Already supplemented and revised: “A total of 120 feather samples were included in this study, representing a minimum of 5 Scaly-sided Mergansers and 3 Common Mergansers.”

3. Line 232 – Common Merganser, the C in Common should be capitalized.

Already corrected.

---

## [Decision Letter · Decision Letter 3]

30 Dec 2025

Dear Dr. Zhao,

Thank you for submitting your manuscript to PLOS ONE. After careful consideration, we feel that it has merit but does not fully meet PLOS ONE’s publication criteria as it currently stands. Therefore, we invite you to submit a revised version of the manuscript that addresses the points raised during the review process.

We look forward to receiving your revised manuscript.

Kind regards,

Tianwen Wang, Ph.D.

Academic Editor

PLOS One

Journal Requirements:

Additional Editor Comments:

Reformat all the tables.

Reviewer's Responses to Questions

**Comments to the Author**

Reviewer #4: All comments have been addressed

Reviewer #5: All comments have been addressed

2. Is the manuscript technically sound, and do the data support the conclusions?

Reviewer #4: Yes

Reviewer #5: Yes

3. Has the statistical analysis been performed appropriately and rigorously?

Reviewer #4: Yes

Reviewer #5: Yes

4. Have the authors made all data underlying the findings in their manuscript fully available?

Reviewer #4: Yes

Reviewer #5: No

5. Is the manuscript presented in an intelligible fashion and written in standard English?

Reviewer #4: Yes

Reviewer #5: Yes

Reviewer #4: This study aimed to differentiate between the Scaly-sided Merganser and the Common Merganser by comparing the microstructures of their contour, rectrices, and down feathers. Their findings suggested that barbule length, node number, and internode distance in down feathers have the potential to be key morphological markers for species differentiation. The manuscript has been reviewed by experts, and the authors have rigorously revised it three times. In my view, the revised manuscript holds significant value and merits publication. I have only minor comments as follows.

Lines 89-90: The sentence “A group of more than 20 Scaly-sided Mergansers was observed through 90 binoculars during the moulting period.” is not precise enough. As far as I know, for the Scaly-sided Mergansers, at least adult males do not molt during the overwintering period, but during the breeding period when females hatch.

Line 130: Please replace “320X to 2600X” with “320× to 2600×”.

Lines 167-168: The table 3 is redundant, should be deleted. Please include the information of table 3 in the contents of results section.

Discussion section: I recommend adding a brief discussion on why contour feathers, rectrices, and down feathers vary in their effectiveness for species identification.

Reviewer #5: This article used scanning electron microscopy to observe the microstructure of the outline feathers, tail feathers, and down feathers of Scaly sided Merganser( Mergus squamatus ) and the Common merganser (M. merger), evaluated the differences between species and determined the potential of these microstructural features as reliable indicators for distinguishing species. The research work has reference value for species identification. The author responded to the reviewer's comments and made revisions in the manuscript. But there are still some areas in the paper that are worth improving.

1. Please clarify the contour feathers are flight feathers or coverts?

2. Is there an intraspecific difference in the ultrastructure of the three types of feathers of the same species?

3. Reference 5: The format of capitalization for scientific names in the titles of references must be consistent with the format of other references.

4. Reference 18: “cairina to mergus”中的“cairina” “mergus” should be italicized.

**Do you want your identity to be public for this peer review?** For information about this choice, including consent withdrawal, please see our Privacy Policy

Reviewer #4: No

Reviewer #5: No

---

## [Author Response · Author response to Decision Letter 4]

8 Jan 2026

Response to Reviewers

Reviewer #4:

1. Lines 89-90: The sentence “A group of more than 20 Scaly-sided Mergansers was observed through binoculars during the moulting period.” is not precise enough. As far as I know, for the Scaly-sided Mergansers, at least adult males do not molt during the overwintering period, but during the breeding period when females hatch.

We sincerely appreciate the reviewer for pointing out this critical oversight, which improves the scientific rigor of our manuscript. We have revised the inaccurate description in the Materials and Methods section (Lines 89–90), correcting “during the moulting period” to “during the field survey”.

2. Line 130: Please replace “320X to 2600X” with “320× to 2600×”.

Already corrected.

3. Lines 167-168: The table 3 is redundant, should be deleted. Please include the information of table 3 in the contents of results section.

Thank you very much for your valuable suggestions. Tables 3, 5, and 7 in the paper constitute a set of corresponding analytical results, which respectively focus on the stepwise discriminant analyses of the two species for contour feathers, rectrices, and down feathers. Removing only Table 3 might disrupt the coherence of the overall structure of the paper. Therefore, we intend to retain Table 3, and we would appreciate it if you could reconsider this arrangement.

4. Discussion section: I recommend adding a brief discussion on why contour feathers, rectrices, and down feathers vary in their effectiveness for species identification.

We have incorporated the recommended discussion in the revised manuscript (Lines 254–259), as follows: This discrepancy in taxonomic resolution across feather types presumably reflects contrasting functional constraints. Contour feathers and rectrices are strongly molded by ecological demands such as flight and display, predisposing them to convergent evolution [24]. Down feathers, specialized primarily for thermoregulation, experience relaxed selective pressure, retain phylogenetically conserved microstructures, and preserve greater species-specific information [25].

Reviewer #5:

1. Please clarify the contour feathers are flight feathers or coverts?

Thank you for your question. As noted in the Sample Preparation section (Lines 111), the contour feathers in this study specifically refer to flank contour feathers, which are neither flight feathers nor coverts in the strict sense.

2. Is there an intraspecific difference in the ultrastructure of the three types of feathers of the same species?

Thank you for this insightful question. As stated in lines 171–173, owing to the limited and poorly documented sample size (number of individuals unknown), we were unable to examine intraspecific variation in the ultrastructure of the three feather types within either the Scaly-sided Merganser or the Common Merganser. This aspect will be addressed once additional specimens of known provenance become available.

3. Reference 5: The format of capitalization for scientific names in the titles of references must be consistent with the format of other references.

Already corrected.

4. Reference 18: “cairina to mergus”中的“cairina” “mergus” should be italicized.

Already corrected.

---

## [Editor Report · Decision Letter 4]

11 Jan 2026

Differences between Scaly-sided Merganser (Mergus squamatus) and Common Merganser (M. merganser) feather microstructure

PONE-D-25-04972R4

Dear Dr. Zhao,

We’re pleased to inform you that your manuscript has been judged scientifically suitable for publication and will be formally accepted for publication once it meets all outstanding technical requirements.

Kind regards,

Tianwen Wang, Ph.D.

Academic Editor

PLOS One
---

## [Editor Report · Acceptance letter]

PONE-D-25-04972R4

PLOS One

Dear Dr. Zhao,

I'm pleased to inform you that your manuscript has been deemed suitable for publication in PLOS One. Congratulations! Your manuscript is now being handed over to our production team.

Kind regards,

on behalf of

Dr. Tianwen Wang

Academic Editor

PLOS One